# Sirtuin1 Mediates the Protective Effects of Echinacoside against Sepsis-Induced Acute Lung Injury via Regulating the NOX4-Nrf2 Axis

**DOI:** 10.3390/antiox12111925

**Published:** 2023-10-29

**Authors:** Weixi Xie, Lang Deng, Miao Lin, Xiaoting Huang, Rui Qian, Dayan Xiong, Wei Liu, Siyuan Tang

**Affiliations:** Xiangya Nursing School, Central South University, Changsha 410013, China; 18508450233@163.com (W.X.); dengl036@163.com (L.D.); linm23508@gmail.com (M.L.); huangxiaoting@csu.edu.cn (X.H.); qianrui521958@outlook.com (R.Q.); xiongdayan@126.com (D.X.)

**Keywords:** echinacoside, SIRT1, oxidative stress, acute lung injury, NOX4, Nrf2

## Abstract

Currently, the treatment for sepsis-induced acute lung injury mainly involves mechanical ventilation with limited use of drugs, highlighting the urgent need for new therapeutic options. As a pivotal aspect of acute lung injury, the pathologic activation and apoptosis of endothelial cells related to oxidative stress play a crucial role in disease progression, with NOX4 and Nrf2 being important targets in regulating ROS production and clearance. Echinacoside, extracted from the traditional Chinese herbal plant *Cistanche deserticola*, possesses diverse biological activities. However, its role in sepsis-induced acute lung injury remains unexplored. Moreover, although some studies have demonstrated the regulation of NOX4 expression by SIRT1, the specific mechanisms are yet to be elucidated. Therefore, this study aimed to investigate the effects of echinacoside on sepsis-induced acute lung injury and oxidative stress in mice and to explore the intricate regulatory mechanism of SIRT1 on NOX4. We found that echinacoside inhibited sepsis-induced acute lung injury and oxidative stress while preserving endothelial function. In vitro experiments demonstrated that echinacoside activated SIRT1 and promoted its expression. The activated SIRT1 was competitively bound to p22 phox, inhibiting the activation of NOX4 and facilitating the ubiquitination and degradation of NOX4. Additionally, SIRT1 deacetylated Nrf2, promoting the downstream expression of antioxidant enzymes, thus enhancing the NOX4-Nrf2 axis and mitigating oxidative stress-induced endothelial cell pathologic activation and mitochondrial pathway apoptosis. The SIRT1-mediated anti-inflammatory and antioxidant effects of echinacoside were validated in vivo. Consequently, the SIRT1-regulated NOX4-Nrf2 axis may represent a crucial target for echinacoside in the treatment of sepsis-induced acute lung injury.

## 1. Introduction

Sepsis was listed as a global health priority by the World Health Organization and the World Health Assembly in 2017 [1]. It has a high clinical mortality rate and affects organs throughout the body [2]. As the most susceptible organ to sepsis, the lungs suffer from acute lung injury (ALI), a common clinical emergency with clinical symptoms ranging in severity from acute dyspnea to end-stage respiratory failure or acute respiratory distress syndrome (ARDS) [3]. The exact pathogenesis of ALI is unknown. Currently, there are no effective therapeutic drugs for ALI/ARDS, and mechanical ventilation remains the primary clinical treatment, but its complications cannot be neglected [4]. The mortality rate for ARDS ranges from 35% to 46%, seriously threatening patient survival [5]. Therefore, there is an urgent need to discover effective drugs that improve the prognosis and quality of life of patients with ALI.

It has been found that endothelial cell activation and injury lead to structural and functional abnormalities of endothelial cells, which are important pathological features of ALI [6]. When sepsis induces an increase in endotoxin, the protective effect of the endogenous anti-inflammatory system in pulmonary endothelial cells is depleted, resulting in the pathological activation of endothelial cells [7]. This activation is characterized by increased expression of adhesion molecules, inflammatory factors, and chemokines, which further promote the recruitment, adhesion, and activation of inflammatory cells [8]. Additionally, endotoxin can directly induce endothelial cell damage and apoptosis, leading to microcirculatory disorders and the worsening of the condition [9]. Therefore, alleviating endothelial cell activation and injury is crucial for the treatment of ALI.

ROS (reactive oxygen species) refers to a collective term for highly reactive substances composed of oxygen that can be found in the body or natural environment [10]. When pathogenic factors are present, the levels of ROS increase dramatically, leading to severe damage to the structure and function of cells, a condition known as oxidative stress [11,12,13]. Growing evidence suggests that oxidative stress is involved in the pathogenesis of ALI, including the pathological activation and injury of endothelial cells [14]. Furthermore, inhibition of oxidative stress has been shown to effectively protect against ALI [15]. The NOX4-Nrf2 axis is an important signaling pathway involved in regulating the levels of reactive oxygen species (ROS) within cells. In this pathway, NOX4 (NADPH oxidase 4) is an enzyme that generates ROS, while Nrf2 (nuclear factor erythroid 2-related factor 2) is a transcription factor that regulates antioxidant responses and cellular defense mechanisms [16,17,18]. Under the influence of LPS, endothelial cells experience disruption in the oxidative balance of the NOX4-Nrf2 axis, leading to an exacerbation of oxidative stress [19,20,21,22]. Therefore, further research on the regulatory mechanisms and functions of the NOX4-Nrf2 axis can provide new targets and strategies for the treatment of ALI.

Sirtuin1 (SIRT1), as a cellular nicotinamide adenine dinucleotide (NAD+)-dependent deacetylase, plays multiple roles in cellular biological processes such as cell longevity, aging, proliferation, DNA repair, apoptosis, inflammation, and cellular metabolism [23,24,25]. Extensive research indicates that SIRT1 can serve as a novel therapeutic strategy for treating ALI and is a critical antioxidant gene [26,27]. SIRT1 has been found to regulate NOX4 and Nrf2 [28,29], but there is currently no research report on the specific regulatory mechanism of SIRT1 on NOX4. Therefore, this study aims to investigate in depth the impact of SIRT1 on NOX4. SIRT1 levels are suppressed in sepsis mice compared to normal mice [30]. Moreover, overexpression of SIRT1 was reported to inhibit LPS-induced inflammatory gene expression [31], and resveratrol, an activator of SIRT1, has been demonstrated to attenuate sepsis-induced cardiac, liver, kidney, and lung injuries by inhibiting oxidative stress [32,33,34]. Although the significant role of SIRT1 in the ALI process has been well established, its exact mechanisms of action remain incompletely understood, and there is limited research on small molecules targeting SIRT1 in ALI treatment. Therefore, there is an urgent need to develop therapeutic drug strategies targeting SIRT1 for ALI treatment.

Echinacoside, extracted and isolated from the traditional Chinese herbal medicine *Cistanche salsa*, is one of its active components. Studies have demonstrated that echinacoside possesses notable immunomodulatory, antioxidant, anti-allergic, anti-inflammatory, and hypoglycemic properties [35,36,37,38]. It has been reported that echinacoside can promote the expression of SIRT1, but the specific effects and mechanisms of echinacoside on ALI have not been investigated [39,40]. Thus, the objective of this study was to explore the in vivo effects of echinacoside mediated by the SIRT1-regulated NOX4-Nrf2 axis on ALI induced by sepsis, as well as its in vitro effects on endothelial cells stimulated by LPS. Additionally, this study aimed to delve into the regulatory mechanisms of SIRT1 on the NOX4-Nrf2 axis.

## 2. Materials and Methods

### 2.1. Experimental Animals

C57BL/6 mice (6–8 weeks old) were housed under specific pathogen-free (SPF) conditions with adequate food and water supply. All animals were obtained from the Animal Center of Central South University. This study was conducted following the principles of animal welfare and ethics. It has been approved by the Laboratory Animal Welfare and Ethics Committee of Xiangya Hospital, Central South University (Approval No. 2022020222).

To investigate the therapeutic effect of echinacoside, a vehicle solution was prepared by diluting echinacoside (Selleck, Houston, TX, USA in 0.9% saline containing 40% PEG300 (Macklin, Shanghai, China), 5% dimethyl sulfoxide (DMSO) (Macklin, China), and 5% Tween80 (Macklin, China) (*v*/*v*/*v*). The mice were randomly divided into five groups and received intraperitoneal (i.p.) injections of echinacoside at doses of 1 mg/kg, 5 mg/kg, 25 mg/kg, or an equivalent amount of the vehicle. Two hours after administering pentobarbital sodium anesthesia, CLP was performed to induce sepsis-induced ALI.

To investigate the molecular mechanism of echinacoside, we selected the dose that showed the most favorable therapeutic effect on sepsis-induced ALI for further investigation. The mice were randomly divided into four groups and received injections of 25 mg/kg echinacoside, 25 mg/kg echinacoside + 10 mg/kg EX527 (a specific SIRT1 inhibitor), or an equivalent amount of the vehicle. Two hours after administering echinacoside, CLP was performed to induce sepsis-induced ALI. EX527 was administered intraperitoneally 3 h before the injection of echinacoside. Lung tissue and bronchoalveolar lavage fluid (BALF) were collected 24 h after CLP.

### 2.2. Histological Analysis

The right middle lung of mice was fixed in 4% paraformaldehyde (Cat# G1101, Servicebio, Wuhan, China) for 24 h. It was then sequentially immersed in 50% and 70% ethanol for 120 min each, 80% and 90% ethanol for 60 min each, 95% ethanol for two 45 min intervals, 100% ethanol for two 30 min intervals, and xylene for two 30 min intervals. The tissue was placed in a culture dish, where melted wax was added for rapid solidification. After embedding, the tissue was cut into 5 μm sections, transferred onto slides, dried, and stained with hematoxylin-eosin.

### 2.3. Immunohistochemistry

The sections were immersed in 3% hydrogen peroxide for 15 min and then subjected to antigen repair by treating them with distilled water at 95 °C for 15 min. Subsequently, the sections were blocked with goat serum and incubated overnight at 4 °C with intercellular adhesion molecule-1 antibody (1:100, Cat# 10020-1-AP, Proteintech, Wuhan, China). After washing the samples three times with PBS, they were incubated with Goat Anti-Rabbit Immunoglobulin G Monoclonal Antibody (1:500, SAB, Nanjing, China) at room temperature for 1 h. Finally, the samples were visualized using 0.05% diaminobenzidine.

### 2.4. Lung Wet to Dry (W/D) Ratio

Lung tissue was removed and weighed on a precision electronic scale (BSA224S-CW; sartorius, Göttingen, Germany), then placed in an oven and baked at 56 °C for 48 h until a constant weight was obtained as dry weight. The W/D ratio was calculated to evaluate the degree of pulmonary edema.

### 2.5. MPO Activity Assay

MPO activity was determined by MPO assay kits (Cat# A044-1-1, Nanjing Jiancheng Bioengineering Institute, Nanjing, China) in lung tissues.

### 2.6. Measurement of MDA, GSH and SOD Levels in Lung Tissues

After euthanizing all the mice, the right lung was excised. Lung tissues were homogenized and dissolved in extraction buffer for the analysis of MDA (malondialdehyde), SOD (superoxide dismutase), and GSH (glutathione) content. The MDA content, representing lipid peroxidation levels in lung tissue, was assessed using commercially available assay kits (Cat# A003-1-2, Nanjing Jiancheng Bioengineering Institute, Nanjing, China) following the manufacturer’s instructions. Furthermore, the activity of antioxidant enzymes in lung tissues, including SOD and GSH, was measured according to the respective manufacturer’s instructions (Cat# GSH: A005-1-2, SOD: Cat# A001-3-2, Nanjing Jiancheng Bioengineering Institute, Nanjing, China).

### 2.7. Bronchoalveolar Lavage Fluid (BALF) Collection and Cell Counts

Mice were euthanized, and the right lungs were lavaged twice with 1 mL of ice-cold PBS. The collected fluid was centrifuged at 4 °C for 10 min at 1500 rpm. The resulting pellet was resuspended in 100 μL of PBS, and the cells were lysed using ACK lysate (Cat# R1010, Solarbio, Beijing, China) for 5 min. Afterward, the cells were washed twice with PBS, and the supernatant was collected and stored at −80 °C for further experiments. Neutrophils and macrophages were counted using a hemocytometer and identified by Wright-Giemsa staining (Cat# G1020, Solarbio, China) under a microscope (Nikon Ti-s, Tokyo, Japan).

### 2.8. Enzyme-Linked Immunosorbent Assay (ELISA)

The concentrations of tumor necrosis factor-alpha (TNF-α) and interleukin-1 beta (IL-1β) in bronchoalveolar lavage fluid (BALF) were determined using ELISA kits (Cat# TNF-α: 88-7324; IL-1β: 88-7013; Invitrogen, Thermo Fisher Scientific, Waltham, MA, USA). The optical density of the samples at 450 nm was measured and compared to the standard curve provided by the kits to quantify the levels of TNF-α and IL-1β.

### 2.9. Cell Culture

Human umbilical vein endothelial cells (HUVECs) were obtained from ATCC and cultured in low-glucose DMEM medium (Cat# PM150220, Procell, Wuhan, China) supplemented with 10% fetal bovine serum (Cat# 0025, ScienCell Research Laboratories, San Diego, CA, USA) and 1% penicillin/streptomycin solution (Cat# 0503, ScienCell Research Laboratories, USA).

### 2.10. ROS Levels and Mitochondrial ROS Levels

ROS levels were measured using H2DCFDA (50 μM, Cat# D399, Invitrogen™, Waltham, MA, USA), and mitochondrial ROS levels were measured using Mitosox (5 μM, Cat#: M36008, Invitrogen, USA), following the manufacturer’s protocol. Briefly, cells were washed twice with cold PBS and resuspended in serum-free DMEM medium at a concentration of 1 × 106 cells/mL. Subsequently, the cells were stained with H2DCFDA dye or Mitosox dye and incubated for 30 min at 37 °C, protected from light. The results were visualized and captured using fluorescence microscopy (Nikon Ti-s, Tokyo, Japan).

### 2.11. RNA Extraction and Quantitative Real-Time Polymerase Chain Reaction (Q-PCR)

The total RNA from lung tissue or cells was extracted using TRIzol Reagent (Cat# 15596026CN, Thermo Fisher Scientific, USA). Subsequently, cDNA was synthesized using a Reverse Transcription Kit (Cat# K1691, Thermo Fisher Scientific, USA). Quantitative polymerase chain reaction (Q-PCR) was performed using SYBR GREEN (Cat# A6001, Promega, Madison, WI, USA) and the Bio-Rad CFX96 Touch Real-Time PCR Detection System (Bio-Rad, Hercules, CA, USA). The Q-PCR conditions consisted of an initial step at 95 °C for 2 min, followed by 40 cycles of amplification at 95 °C for 3 s and 60 °C for 30 s. A melting curve analysis was performed from 60 °C to 95 °C. The primer sequences: HO-1, F-ACCGCCTTCCTGCTCAACATTG, R-CTCTGACGAAGTGACGCCATCTG; NQO-1, F-GCGAGAAGAGCCCTGATTGTACTG, R-AGCCTCTACAGCAGCCTCCTTC; ICAM-1, F-CGCAGAGGACCTTAACAGTCTACAAC, R-CTTCACAGTTACTTGGCTCCCTTCC; VCAM-1, F-TGATTGGGAGAGACAAAGCAGAAGTG, R-CAATAGCAGCACACGTCAGAACAAC; IL-6, F- CTTCTTGGGACTGATGCTGGTGAC, R-AGGTCTGTTGGGAGTGGTATCCTC; IL-1β, F-TCGCAGCAGCACATCAACAAGAG, R-AGGTCCACGGGAAAGACACAGG; TNF-α, F-GCCTCTTCTCATTCCTGCTTGTGG, R-GTGGTTTGTGAGTGTGAGGGTCTG. β-actin, F-CCTGCGACTTCAACAGCAAC, R-TGGGATAGGGCCTCTCTTGC; SIRT1, F-ATGCCAGAGTCCAAGTTTAGAAGAACC, R-AAATCCAGATCCTCCAGCACATTCG.

### 2.12. Western Blotting

Total lung tissue or cells were extracted using RIPA lysis buffer (Cat# R0020, Solarbio, China). The protein concentrations were detected using the bicinchoninic acid (BCA) protocol. Ten percent of the proteins were separated by sodium dodecyl sulfate-polyacrylamide gel electrophoresis (SDS-PAGE) and transferred onto PVDF membranes (Cat# 1620177, Bio-Rad, USA). The membranes were blocked with 5% (*w*/*v*) skim milk in TBS buffer containing 0.1% (*v*/*v*) Tween 20 (TBST). The membranes were then incubated with specific primary antibodies overnight at 4 °C, including β-actin antibody (1:5000, Cat# 66009-1-Ig, Proteintech, China), VCAM-1 antibody (1:1000, Cat# bs-0920R, Bioss, Boston, MA, USA), ICAM-1 antibody (1:1000, Cat# 10020-1-AP, Proteintech, China), SIRT1 polyclonal antibody (1:1000, CAT# 13161-1-AP, Proteintech, China), HO-1 antibody (1:2000, Cat# ab189491, Abcam, Cambridge, UK), NQO-1 antibody (1:2000, CAT# ab80588, Abcam, UK), NOX4 antibody (1:1000, Cat# 14347-1-AP, Proteintech, China), p65 antibody (1:1000, Cat# 8242T, CST, Danvers, MA, USA), Pho-p65 antibody (1:1000, Cat# 3033T, CST, USA), IκB α antibody (1:1000, Cat# ab32518, Abcam, UK), Pho-IκB α antibody (1:10,000, Cat# ab133462, Abcam, UK), Pho-JNK antibody (1:1000, Cat# 9251S, CST, USA), JNK antibody (1:1000, Cat# 9252T, CST, USA), ERK1/2 antibody (1:1000, Cat# 4695T, CST, USA), Pho-ERK1/2 antibody (1:1000, Cat# 4370T, CST, USA), Pho-p38 antibody (1:1000, Cat# 28796-1-AP, Proteintech, China), p38 antibody (1:1000, Cat# 14064-1-AP, Proteintech, China), Nrf2 antibody (1:1000, Cat# 16396-1-AP, Proteintech, China), Ubquination antibody (1:500, Cat# A162, Abclonal, Wuhan, China) and p22 phox antibody (1:1000, Cat# A10694, ABclonal, China). After washing with TBST, the membranes were incubated with horseradish peroxidase-conjugated secondary antibodies, either goat anti-rabbit immunoglobulin G monoclonal antibody (1:5000, Cat# L3012, SAB, China) or goat anti-mouse immunoglobulin G monoclonal antibody (1:5000, Cat# L3032, SAB, China), for 2 h at room temperature. The protein bands were visualized using Luminata™ Crescendo chemiluminescent horseradish peroxidase substrate (Millipore, Burlington, VT, USA) and scanned using a GeneGnome XRQ imager (Syngene, Cambridge, UK).

### 2.13. Mitochondrial Membrane Potential (ΔΨm) Measurement

Changes in mitochondrial membrane potential (ΔΨm, MMP) were assessed using flow cytometry (BD LSRFortessa, Franklin Lakes, NJ, USA). The JC-1 indicator reagent was employed to measure MMP in HUVECS following exposure to LPS (1 μg/mL). Cells were cultured on 12-well plates and confocal dishes and subjected to various experimental treatments. After 24 h of LPS exposure, cells were harvested and incubated with JC-1 (5 μM, Cat# C2005, Beyotime, Shanghai, China) at 37 °C for 20 min. Subsequently, the labeled cells were washed with buffer for imaging and recording.

### 2.14. Immunofluorescence

Cells were fixed in 4% paraformaldehyde for 15 min, permeabilized with 0.5% Triton X-100 for 20 min, and closed with goat serum for 30 min. Then, they were incubated overnight at 4 °C with Cytochrome C (1:100, Cat# 1000345-2, Abcam, UK), SIRT1 (1:100, CAT# 13161-1-AP, Proteintech, China), NOX4 (1:100, CAT# 67681-1-Ig, Proteintech, China), and p22 phox antibody (1:100, Cat# A10694, ABclonal, China). After PBS washing, the cells were incubated with CoraLite488-conjugated Goat Anti-Rabbit IgG(H+L) (1:100, Cat# SA00013-2, Proteintech, China) or CoraLite594—conjugated Goat Anti-Mouse IgG(H+L) (1:100, Cat# SA00013-3, Proteintech, China) at 37 °C for 1 h and observed under fluorescence microscopy (Zeiss Apotome, Oberkochen, Germany).

### 2.15. Cell Apoptosis Assay

Flow cytometry was performed to assess cell apoptosis. Cells were pretreated with echinacoside for 1 h and LPS (1 μg/mL) for another 24 h. The Annexin V-FITC Apoptosis Detection Kit (Cat# C1062L, Beyotime, China) was used for apoptosis assays according to the manufacturer’s instructions. Briefly, cells were washed twice with cold PBS and resuspended in binding buffer at a concentration of 1 × 106 cells/mL. Subsequently, cells were stained with 3 μL of FITC Annexin V and 5 μL of propidium iodide and incubated at room temperature in the dark for 15 min. Finally, the samples were analyzed using flow cytometry (BD LSRFortessa, Franklin Lakes, NJ, USA), and the data were analyzed using Flowjo 10.8.1 software.

### 2.16. siRNA Transfection

siRNAs specific for SIRT1 (Cat# sc-40986, Santa Cruz, Dallas, TX, USA) or scrambled control (Cat# sc-37007, Santa Cruz, USA) were transfected into HUVECs using Lipofectamine™ RNAiMAX (Cat# 13778150, Invitrogen, USA). Knockdown efficiency was determined by Western blotting.

### 2.17. Co-Immunoprecipitation

To investigate the interaction between NOX4 and p22 phox, Nrf2 and ac-lysine, ubquintination and NOX4, the following procedure was performed. Cells were washed three times with PBS and then lysed on ice for 1 h using lysis buffer containing complete protease inhibitor PMSF (Cat# P0100, Solarbio, China). The lysates were collected and centrifuged at 12,000 rpm for 10 min at 4 °C. The resulting supernatant was incubated with Dynabeads™ Protein G (Cat# 10004D, Invitrogen, USA) for 3 h at 4 °C, followed by centrifugation. The Dynabeads were then isolated using DynaMag™-2 (Cat# 12321D, Invitrogen, USA). The supernatant was incubated overnight at 4 °C with SIRT1, NOX4, or ac-lysine. The beads were subsequently washed three times with lysis buffer, and the Dynabeads were separated using DynaMag™-2. Immunoprecipitated proteins were analyzed using Western blotting.

### 2.18. Statistical Analysis

All data were expressed as means ± standard deviations and analyzed using GraphPad Prism 9.0 software. Firstly, we performed a Shapiro–Wilk test to assess the normality of the data distribution. For data that followed a normal distribution, one-way analysis of variance (ANOVA) was used for intergroup statistical comparisons, followed by the Tukey test for further pairwise group comparisons. For data that did not follow a normal distribution, we employed the Kruskal–Wallis test. Survival analysis was conducted using the log-rank test. Statistical significance was considered when the *p*-value was less than 0.05.

## 3. Results

### 3.1. Echinacoside Attenuated Sepsis-Induced Acute Lung Injury in Mice and Preserved Endothelial Cell Function

As a common animal model for ALI, CLP rapidly induces sepsis, thereby triggering a cascade of inflammatory reactions caused by endotoxins in the body, leading to an inflammatory storm and subsequently causing lung injury [41]. Therefore, we chose to administer intraperitoneal injections of echinacoside at doses of 1 mg/kg, 5 mg/kg, and 25 mg/kg to observe the protective effects against sepsis-induced acute lung injury. The HE results revealed that the reduction of the echinacoside concentration gradient alleviated the damage to alveolar structural integrity, enlargement of the alveolar cavity and interstitium, as well as an increase in the inflammatory cell population caused by sepsis (Figure 1A). Additionally, echinacoside mitigated the elevated activity of MPO induced by sepsis, the increase of the lung W/D ratio, as well as the elevated levels of total protein in BALF and the increased numbers of macrophages and neutrophils (Figure 1B–F). TNF-α, IL-1β, and IL-6 are markers of an inflammatory response. The levels of TNF-α and IL-1β in the BALF were lower in the echinacoside group compared to the CLP group (Figure 1G,H). Additionally, the levels of TNF-α, IL-1β, and IL-6 in the lung tissue homogenate were also reduced in the echinacoside group compared to the CLP group (Figure 1I–K).

Endothelial cells, which account for 20–30% of pulmonary parenchymal cells, play a vital role in the pathophysiological processes of various lung diseases [42]. VCAM-1 and ICAM-1, as markers of endothelial cell pathological activation, are important indicators of endothelial cell function. Immunohistochemistry revealed that echinacoside alleviated the upregulation of VCAM-1 expression in endothelial cells induced by sepsis (Figure 1L). qPCR and Western blotting demonstrated that the mRNA and protein levels of VCAM-1 and ICAM-1, induced by sepsis, were elevated, but this elevation was mitigated by echinacoside treatment (Figure 1M–Q). To summarize, echinacoside exhibited a protective effect on endothelial cell function and mitigated sepsis-induced ALI.

### 3.2. Echinacoside Alleviated Sepsis-Induced Lung Oxidative Stress in Mice

Oxidative stress is considered to play a significant role in the pathogenesis of ALI [15]. Therefore, we investigated the effect of echinacoside on sepsis-induced lung oxidative stress. DHE staining demonstrated that the reduction of the echinacoside concentration gradient mitigated the elevated levels of lung oxidative stress induced by sepsis (Figure 2A). Similarly, echinacoside reversed the increased levels of MDA and decreased levels of reduced GSH and SOD activity in the lungs induced by sepsis (Figure 2B–D). Moreover, echinacoside reduced the elevated levels of ROS in the BALF induced by sepsis (Figure 2E). NOX4 and Nrf2 are key targets involved in ROS production and clearance in the body’s antioxidant system [20,21]. Therefore, we investigated whether echinacoside exerted its antioxidant effects through the NOX4-Nrf2 axis. The qPCR results revealed that ECH treatment increased the levels of HO-1 and NQO-1, as the downstream targets of Nrf2 (Figure 2F,G). Western blotting showed that, compared to the CLP group, echinacoside treatment significantly reduced the level of NOX4 and markedly increased the levels of HO-1 and NQO-1 (Figure 2H–K). In conclusion, echinacoside attenuated sepsis-induced lung oxidative stress in mice, and this effect may be associated with the NOX4-Nrf2 axis.

### 3.3. Echinacoside Mitigated the Pathological Activation of Endothelial Cells by Inhibiting LPS-Activited the NF-κB and MAPK Signaling Pathways

The NF-κB and MAPK signaling pathways are crucial pathways involved in the pathological activation of endothelial cells, which are activated by LPS [43,44]. Therefore, we investigated whether echinacoside could alleviate the activation of NF-κB and MAPK signaling pathways induced by LPS. The results revealed that the inhibition of NF-κB and MAPK signaling pathways was observed with the reduction of the echinacoside concentration gradient compared to the LPS group (Figure 3A–G). Additionally, qPCR and Western blotting analysis demonstrated that the elevation of VCAM-1 and ICAM-1 mRNA and protein levels induced by LPS was blocked by echinacoside (Figure 3H–L). In conclusion, echinacoside can mitigate the pathological activation of endothelial cells by inhibiting the NF-κB and MAPK signaling pathways activated by LPS.

### 3.4. Echinacoside Enhanced the NOX4-Nrf2 Axis, Thereby Attenuating LPS-Induced Apoptosis via the Mitochondrial Pathway

Given the in vivo enhancement of the NOX4-Nrf2 axis by echinacoside, we also investigated the effect of ECH on the NOX4-Nrf2 axis in vitro. The results indicated that echinacoside reduced the upregulation of NOX4 induced by LPS and promoted the expression of HO-1 and NQO-1 (Figure 4A–F), thereby reducing ROS levels (Figure 4G–I). The apoptosis mediated by the ROS-regulated mitochondrial pathway is a major cause of endothelial cell injury. Given the regulatory effect of echinacoside on ROS, we next investigated whether echinacoside could reduce apoptosis through the mitochondrial pathway in endothelial cells. Mitosox staining indicated that echinacoside reversed the elevated levels of mitochondrial ROS induced by LPS (Figure 4J). JC-1 staining reflected that echinacoside prevented the increase in mitochondrial membrane potential induced by LPS in endothelial cells (Figure 4K,L), and it reduced the leakage of cytochrome C from mitochondria induced by LPS (Figure 4M), thereby reducing apoptosis through the mitochondrial pathway (Figure 4N). In conclusion, echinacoside alleviated ROS-mediated endothelial cell apoptosis through the NOX4-Nrf2 axis in the mitochondrial pathway.

### 3.5. SIRT1 Mediated the Protective Effects of Echinacoside against LPS-Induced Pathological Activation of Endothelial Cells

Due to the regulatory effects of echinacoside on both NOX4 and Nrf2, it is speculated that echinacoside may have other upstream targets. It has been reported that SIRT1, as a cytosolic nicotinamide adenine dinucleotide (NAD+)-dependent deacetylase, regulates NOX4 and Nrf2 [28,29]. We observed an increase in the mRNA and protein levels of SIRT1 both in vivo and in vitro upon echinacoside treatment (Figure 5A–G). Furthermore, we found that the inhibitory effects of echinacoside on the NF-κB and MAPK signaling pathways were reversed upon SIRT1 siRNA treatment (Figure 5H–N). In the NF-κB signaling pathway, the nuclear translocation of p65 is mainly regulated by its phosphorylation levels, while the transcriptional activity of p65 is regulated by acetylation. As SIRT1 functions as a deacetylase, we hypothesized that ECH could promote the deacetylation of p65, and our results aligned with this hypothesis (Figure 5O,P). Meanwhile, our original discovery indicates that activated SIRT1 can directly interact with p65, potentially inhibiting the nuclear translocation of p65 (Figure 5Q). qPCR and Western blotting results indicated that the inhibitory effects of echinacoside on VCAM-1 and ICAM-1 were also blocked by SIRT1 siRNA treatment (Figure 5R–V). In conclusion, echinacoside promoted the expression of SIRT1, which in turn inhibits the pathological activation of endothelial cells mediated by the NF-κB and MAPK signaling pathways.

### 3.6. SIRT1 Mediated the Protective Effects of Echinacoside on LPS-Induced Mitochondrial Pathway Apoptosis of Endothelial Cells

After treatment with SIRT1 siRNA, the effects of echinacoside on NOX4 and Nrf2 were also reversed (Figure 6A–D). Next, we investigated whether SIRT1 mediated the effects of echinacoside on endothelial cell mitochondrial pathway apoptosis. Under the influence of SIRT1 siRNA, the effects of echinacoside on ROS, mitochondrial ROS, mitochondrial membrane potential, and apoptosis were blocked (Figure 6E–J). In summary, SIRT1 mediates the regulatory effects of echinacoside on the NOX4-Nrf2 axis in endothelial cells and the protective effects of echinacoside against LPS-induced mitochondrial pathway apoptosis.

### 3.7. Activated SIRT1 Inhibits the Activation of NOX4 and Promotes the Ubiquitination Degradation of NOX4

Given the role of SIRT1 in NOX4 and Nrf2, we further investigated the regulatory mechanisms of SIRT1 in NOX4 and Nrf2. Current research on NOX4 and SIRT1 has primarily focused on the activation of SIRT1 regulating the expression of NOX4, with limited studies exploring whether SIRT1 can modulate the activity of NOX4. As the cytoplasmic regulatory subunit, p22 phox is recruited and binds to NOX4 upon activation [45]. Therefore, we chose to observe the activity of NOX4 after 3 h of treatment with echinacoside (where NOX4 expression remains unchanged). The results from immunofluorescence and CO-IP demonstrate that activation of SIRT1 can disrupt the binding between p22 phox and NOX4, possibly due to the competitive binding of activated SIRT1 with p22 phox, and we also discovered the well-known activator of SIRT1, resveratrol (Figure 7A–C). Moreover, our original discovery revealed that the activation of SIRT1 promotes the ubiquitination degradation of NOX4, which may be associated with the inhibition of the binding between NOX4 and p22 phox, thereby preventing the formation of a stable NOX4 complex structure (Figure 7D). Furthermore, the inhibitory effect of activated SIRT1 on the binding between p22 phox and NOX4, as well as the promotion of NOX4 degradation by SIRT1, is reversed in the presence of SIRT1 siRNA (Figure 4F–I and Figure 7D). Regarding the regulatory role of SIRT1 on Nrf2, we observed that activated SIRT1 can deacetylate Nrf2, which may be one of the mechanisms by which SIRT1 promotes the expression of Nrf2 downstream genes HO-1 and NQO-1 (Figure 7E). In conclusion, SIRT1 competitively binds to p22 phox, thereby influencing the interaction between NOX4 and p22 phox, leading to the inhibition of NOX4 activity. Additionally, SIRT1 promotes the ubiquitination and degradation of NOX4, possibly due to the disruption of stable complex formation between NOX4 and p22 phox. Furthermore, activated SIRT1 can enhance the expression of downstream genes of Nrf2 by deacetylating Nrf2.

### 3.8. SIRT1 Regulated the Protective Effects of Echinacoside In Vivo against Sepsis-Induced ALI and the Preservation of Endothelial Cell Function

To investigate the role of SIRT1 in echinacoside-mediated protection against sepsis-induced ALI, a specific inhibitor of SIRT1, EX527, was injected 3 h prior to echinacoside treatment. HE staining indicated that EX527 reversed the protective effect of echinacoside on ALI, which was further validated by MPO levels, W/D ratio of lung, TNF-α, IL-1β, and IL-6 levels in lung tissue, total protein levels, macrophage and neutrophil counts, as well as TNF-α and IL-1β measurements in BALF (Figure 8A–J). Additionally, EX527 reversed the impact of echinacoside on endothelial cell function (Figure 8K–P). In conclusion, SIRT1 mediated the protective effects of echinacoside in mice with sepsis-induced ALI.

### 3.9. SIRT1 Mediates the Antioxidative Effect of Echinacoside In Vivo

DHE staining showed that EX527 blocked the antioxidant effects of echinacoside, which was further validated by measurements of MDA levels, GSH levels, and SOD activity (Figure 9A–D). The regulatory effects of echinacoside on NOX4 and Nrf2 were also reversed by EX527 (Figure 9E–J). In summary, SIRT1 mediates the antioxidant effects of echinacoside in vivo.

## 4. Discussion

Our study provides the first evidence that echinacoside can activate SIRT1 and increase SIRT1 levels both in vitro and in vivo. Activated SIRT1 competes with p22 phox, inhibiting the interaction between NOX4 and p22 phox, thus suppressing NOX4 activity. And the activation of SIRT1 leads to a decrease in the interaction between NOX4 and p22 phox, which affects the stability of NOX4 and promotes its ubiquitination degradation. Additionally, SIRT1 deacetylates Nrf2, promoting the expression of downstream antioxidant enzymes and thereby enhancing the NOX4-Nrf2 axis. SIRT1 also mitigates the pathological activation of endothelial cells by inhibiting the ROS-regulated NF-κB and MAPK signaling pathways. Importantly, we discovered that activated SIRT1 directly binds to p65, inhibiting p65 nuclear translocation, and deacetylates p65, thereby inhibiting p65 nuclear transcription. Similarly, SIRT1 mediates the protective effects of echinacoside in attenuating ROS-mediated endothelial cell mitochondrial apoptosis, thereby alleviating sepsis-induced acute lung injury in mice.

Multiple studies have demonstrated that sepsis-induced ALI is associated with vascular dysfunction [42]. Under pathogen infection, trauma, or widespread inflammation, endothelial cells become hyperpermeable, allowing proteins and fluids to extravasate [3]. The disruption of endothelial barrier function is considered the primary cause of ALI development and is associated with increased mortality [6]. Endothelial cell pathological activation and injury are the main pathological features of endothelial barrier disruption [8,9]. It is important to emphasize that endothelial cell activation is a normal part of the body’s defense mechanisms. Under physiological conditions, endothelial cells play a role in normal physiological functions to resist stress and injury. However, when exposed to dangerous factors, the protective effect of the endogenous anti-inflammatory system in endothelial cells becomes depleted, leading to pathological activation of endothelial cells. This is characterized by increased expression of adhesion molecules, inflammatory factors, and chemokines, which in turn recruit, adhere to, and activate circulating leukocytes on the vascular wall [8]. Additionally, dangerous factors can directly induce endothelial cell injury and apoptosis, disrupting the integrity of the vascular endothelium and leading to microcirculatory disorders, thereby exacerbating the condition [46]. Therefore, reducing endothelial cell pathological activation and injury is crucial for the treatment of ALI. In the CLP-induced ALI animal model, the greatest difference in vascular endothelial permeability occurs between 6 and 24 h after injury [47]. Our results demonstrate that echinacoside significantly inhibits the expression of endothelial adhesion molecules and reduces vascular permeability, and in vitro results show that echinacoside suppresses pathological activation and mitochondrial pathway apoptosis in endothelial cells. Based on these findings, it can be inferred that echinacoside may protect pulmonary vascular endothelial function and alleviate lung injury. These findings suggest that echinacoside has great potential as a therapeutic agent for ALI.

Oxidative stress is considered to play a significant role in the pathogenesis of ALI, and ROS are crucial mediators in the activation and damage of endothelial cells by various pathogenic factors such as LPS [48]. Extensive research has shown that ROS act as enhancers for the activation of the NF-κB signaling pathway by various stimulating factors [49]. The antioxidant NAC can alleviate NF-κB activation and the production of pro-inflammatory cytokines, suggesting that ROS play an important role in the activation of the NF-κB signaling pathway induced by LPS [50]. The MAPK signaling pathway is also influenced by ROS, with ROS mediating the activation of the MAPK signaling pathway [51]. For example, ROS can induce the oxidation of thioredoxin to promote the release of the ASK1 complex and subsequently activate the MAPK signaling pathway [39]. Therefore, combating ROS is beneficial for reducing the activation of endothelial cells. Our results indicate that echinacoside can reduce the levels of intracellular ROS and mitoROS induced by LPS stimulation, thereby alleviating oxidative stress. Additionally, we observed inhibition of the NF-κB and MAPK signaling pathways. Furthermore, excessive ROS can promote cell apoptosis through various pathways, with the mitochondrial pathway being the primary cause. We found that echinacoside treatment can inhibit the decline of mitochondrial membrane potential, alleviate mitochondrial damage, and suppress endothelial cell apoptosis.

In some previous studies, echinacoside was found to inhibit the AKT signaling pathway, increase intracellular ROS levels, and promote apoptosis in endometrial cancer cells at higher concentrations. This finding raised some inconsistencies with the experimental results of our study. To address these differences, a series of experiments were conducted. Firstly, we treated HUVECs with high concentrations of echinacoside (255 μM) and the effective low concentration used in our study (30 μM) to investigate the impact of echinacoside at varying concentrations on intracellular ROS levels. The results showed that high concentrations of echinacoside significantly increased intracellular ROS content in HUVECs and induced apoptosis, while low-concentration echinacoside treatment did not show significant effects. Furthermore, we delved into the mechanisms underlying this phenomenon and found that high-concentration echinacoside altered mitochondrial membrane potential, leading to an increase in mitochondrial ROS levels. This suggests that the high concentration of echinacoside may increase intracellular ROS and promote apoptosis by inducing mitochondrial damage (Appendix A).

Cells possess a large number of antioxidant systems that can to some extent reduce the generation of ROS or increase the clearance of ROS, thereby combating the endothelial cell activation and damage caused by oxidative stress [52,53]. NOX4, as a key factor in ROS production, is regulated at the transcriptional level by the NF-κB and MAPK signaling pathways in response to LPS [54]. Our results have found that echinacoside treatment can upregulate NOX4 levels in both in vivo and in vitro, which may be related to the inhibition of the NF-κB and MAPK signaling pathways by echinacoside. Nrf2, as a key factor in the antioxidant stress response, can bind to antioxidant response elements (AREs) and activate the transcription of antioxidant proteins such as HO-1 or NQO-1, thereby clearing excessive ROS and alleviating oxidative stress-induced damage [55]. It has also been shown that Nrf2 can competitively bind with NF-κB co-transcriptional factor CBP/p300, thus regulating the inflammatory response [56]. In this study, we found that echinacoside can enhance the levels of HO-1 and NQO-1 while attenuating oxidative stress-induced damage. Although compounds targeting Nrf2 or NOX4 have been studied, there are few compounds specifically targeting the NOX4-Nrf2 axis. Therefore, due to its enhancement of the NOX4-Nrf2 axis, echinacoside holds great potential for oxidative stress-involved diseases.

SIRT1 is a nicotinamide adenine dinucleotide (NAD+)-dependent deacetylase, which plays a critical role in preventing or delaying ALI and also functions as an antioxidant (24). It has been reported that the absence of SIRT1 in the liver, pancreas, and brain significantly increases the occurrence of ROS and inflammatory responses [57]. Additionally, resveratrol, an activator of SIRT1, has been shown to alleviate heart, liver, kidney, and lung injuries [32,33,34]. Although the regulation of SIRT1 by echinacoside has been studied, its role in acute lung injury has not been investigated. Our study revealed that echinacoside treatment promoted SIRT1 expression both in vitro and in vivo. The effects of echinacoside on endothelial cell activation and injury caused by oxidative stress were reversed by SIRT1 siRNA. Moreover, the protective effects of echinacoside against sepsis-induced ALI and lung oxidative stress were blocked by EX527, a SIRT1 inhibitor. These findings suggest that echinacoside inhibits oxidative stress and protects endothelial cells through SIRT1 mediation in the context of ALI. Considering the regulatory role of the AKT signaling pathway in NADPH oxidase activity, we also separately examined the effects of different echinacoside concentrations on the AKT signaling pathway. The findings revealed that high concentrations of echinacoside notably inhibited AKT activation, whereas no such effect was observed following low-concentration echinacoside treatment. This suggests that the inhibitory effect of echinacoside on the AKT signaling pathway is concentration-dependent. In our study, we used a concentration of 30 μM echinacoside, which significantly enhanced Sirt1 activity while having no discernible impact on the AKT signaling pathway (Appendix A). Additionally, we discovered that the antioxidant damage-repairing effect of echinacoside decreased significantly after knocking down SIRT1 expression. This implies that SIRT1 plays a crucial role in the biological effects of low-concentration echinacoside; however, further investigation is needed regarding the effects of high-concentration echinacoside on SIRT1. It is worth noting that the SIRT1 activator, Resveratrol, exhibits similar pharmacological activities. Its effects on the cellular oxidative stress response are also bidirectional, with low concentrations displaying antioxidant stress and cell protection effects, while high concentrations promote oxidative stress and cytotoxicity. 

Furthermore, our study found that the effects of echinacoside on NOX4 and Nrf2 were reversed under SIRT1 siRNA treatment, suggesting that the actions of echinacoside on the NOX4-Nrf2 axis are mediated by SIRT1. Therefore, we investigated the mechanisms by which SIRT1 regulates the NOX4-Nrf2 axis. Previous studies have shown that SIRT1 regulates the redox balance in the liver by deacetylating Nrf2, leading to the expression of antioxidant enzymes [58], which is consistent with our findings. However, the regulatory mechanism of SIRT1 on NOX4 remains unclear, as existing research only focuses on the inhibitory effect of increased SIRT1 levels on NOX4 expression. We speculate that the activation of NOX4 may also be regulated by SIRT1. p22 phox acts as a subunit of the NOX enzyme and forms an active NOX enzyme complex when bound to NOX4 [45]. Therefore, we investigated the effect of echinacoside on the interaction between p22 phox and NOX4. The results showed that echinacoside is able to inhibit the activation of NOX4, and this interaction may be associated with the competitive binding of activated SIRT1 to p22 phox. Additionally, it has been reported that the binding of NOX4 and p22 phox promotes the stability of NOX4 [59]. Therefore, we hypothesized that the degradation of NOX4 would occur when activated SIRT1 competitively binds to p22 phox, and our results confirmed this hypothesis. The above results provided the first elucidation of the mechanism by which activated SIRT1 affects the NOX4-Nrf2 pathway, offering new evidence for the future regulation of SIRT1 in disease pathology.

## 5. Conclusions

Our research results demonstrate for the first time that echinacoside is a promising anti-inflammatory and antioxidant agent for the treatment of ALI. Echinacoside exerts its effects through the SIRT1-mediated inhibition of ROS-mediated endothelial cell activation and injury in the NOX4-Nrf2 axis. Mechanistically, activated SIRT1 competitively binds to p22 phox, reducing the interaction between NOX4 and p22 phox, thereby inhibiting NOX4 activity and promoting NOX4 ubiquitination degradation. Additionally, echinacoside deacetylates Nrf2, enhancing the NOX4-Nrf2 axis. Lastly, it is important to note that while the therapeutic effects and mechanisms of echinacoside on ALI have been explored in this study, the significance levels used may not have adequately addressed the issue of multiple testing involving various indicators and outcomes within the study. This limitation could potentially lead to an overestimation of statistical significance. The anti-inflammatory and antioxidant properties of echinacoside warrant further investigation in other inflammation-related diseases and conditions associated with endothelial cell activation and injury.

## Figures and Tables

**Figure 1 antioxidants-12-01925-f001:**
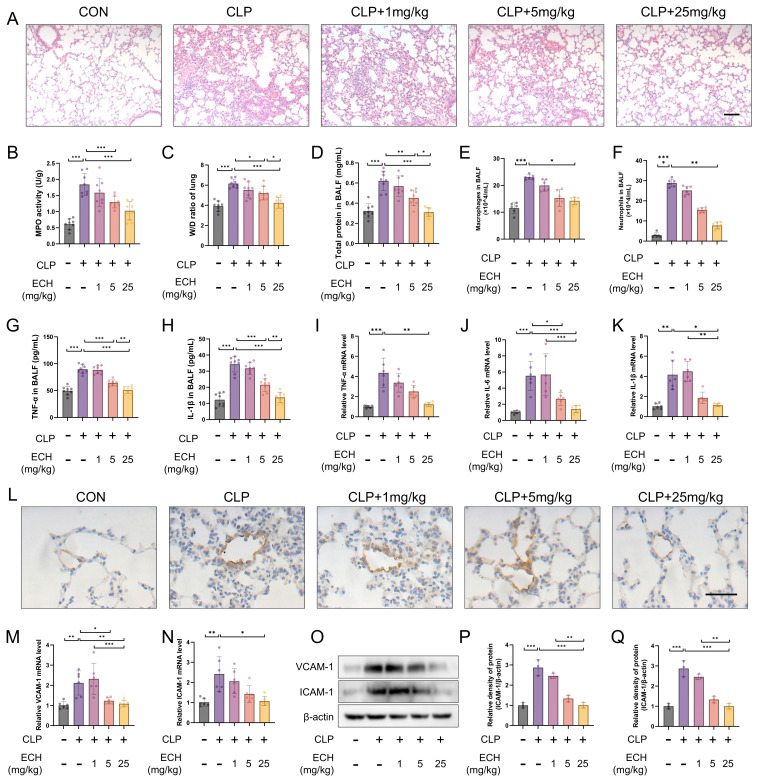
Echinacoside alleviated sepsis-induced acute lung injury and preserved endothelial cell function. Intraperitoneal injections of Ech at doses of 1 mg/kg, 5 mg/kg, and 25 mg/kg were administered during CLP to assess its effects on lung injury and pulmonary endothelial cells. (**A**) Lung morphology was assessed by HE staining. Bars represent 100 μm. The microscope’s magnifications are the same. (**B**) MPO levels were measured using biochemical methods. (**C**) The lung W/D ratio was determined. (**D**) Total protein content in BALF was measured using the BCA method. (**E**,**F**) Gimsa staining was performed to determine macrophage and neutrophil levels in BALF. (**G**,**H**) ELISA was used to measure TNF-α and IL-1β levels in BALF. (**I**–**K**) qPCR was employed to measure the mRNA levels of TNF-α, IL-6, and IL-1β in lung tissue homogenates. (**L**) Immunohistochemistry was used to assess VCAM-1 levels in lung tissue. Bars represent 100 μm. The microscope’s magnifications are the same. (**M**–**Q**) qPCR and Western blotting were used to measure the mRNA and protein levels of VCAM-1 and ICAM-1 in lung tissue. The data are presented as mean ± standard deviation, and all experiments were repeated independently at least three times. (* *p* < 0.05, ** *p* < 0.01, *** *p* < 0.001).

**Figure 2 antioxidants-12-01925-f002:**
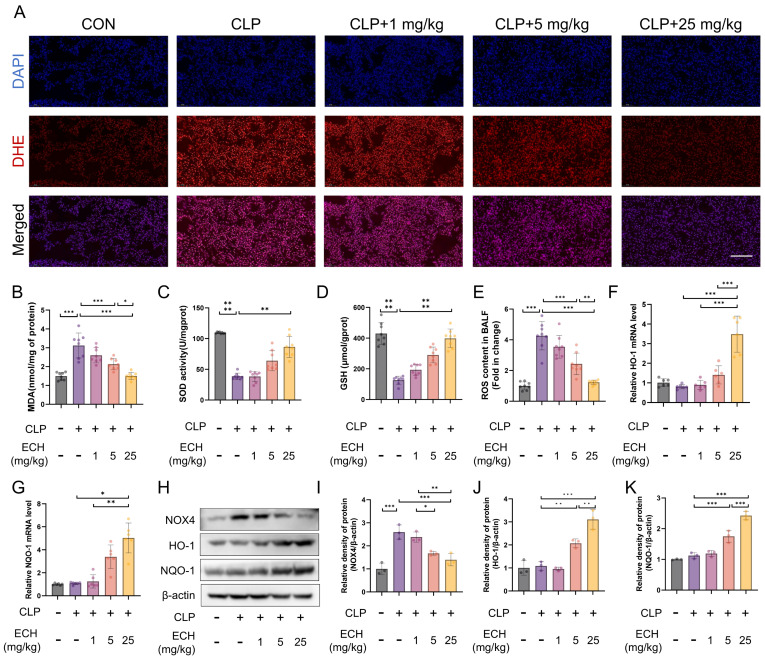
Echinacoside alleviated sepsis-induced lung oxidative stress. (**A**) DHE staining reflects the level of pulmonary oxidative stress. Bars represent 100 μm. The microscope’s magnifications are the same. (**B**–**D**) Biochemical methods were used to measure the levels of MDA and GSH as well as the activity of SOD. (**E**) The H2DCFCDA probe was used to stain ROS in BALF and was analyzed using flow cytometry. (**F**–**K**) qPCR and Western blotting were employed to measure the mRNA levels of HO-1 and NQO-1 and the protein levels of HO-1, NQO-1, and NOX4. The data are presented as mean ± standard deviation, and all experiments were repeated independently at least three times. (* *p* < 0.05, ** *p* < 0.01, *** *p* < 0.001).

**Figure 3 antioxidants-12-01925-f003:**
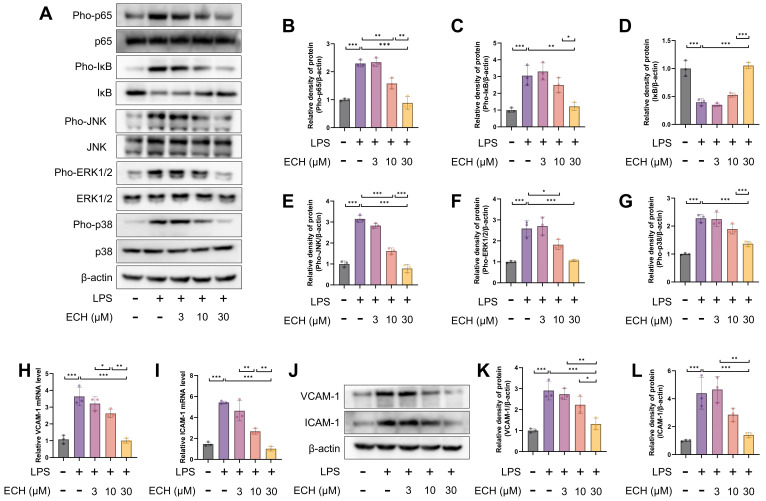
Echinacoside alleviated LPS-induced pathological activation of endothelial cells. (**A**–**G**) Protein levels of Pho-p65, p65, Pho-IκB, IκB, Pho-ERK1/2, ERK1/2, Pho-JNK, JNK, Pho-p38, and p38 were detected by Western blotting. (**H**–**L**) mRNA and protein levels of VCAM-1 and ICAM-1 were measured using quantitative PCR and Western blotting. The data are presented as mean ± standard deviation, and all experiments were repeated independently at least three times. (* *p* < 0.05, ** *p* < 0.01, *** *p* < 0.001).

**Figure 4 antioxidants-12-01925-f004:**
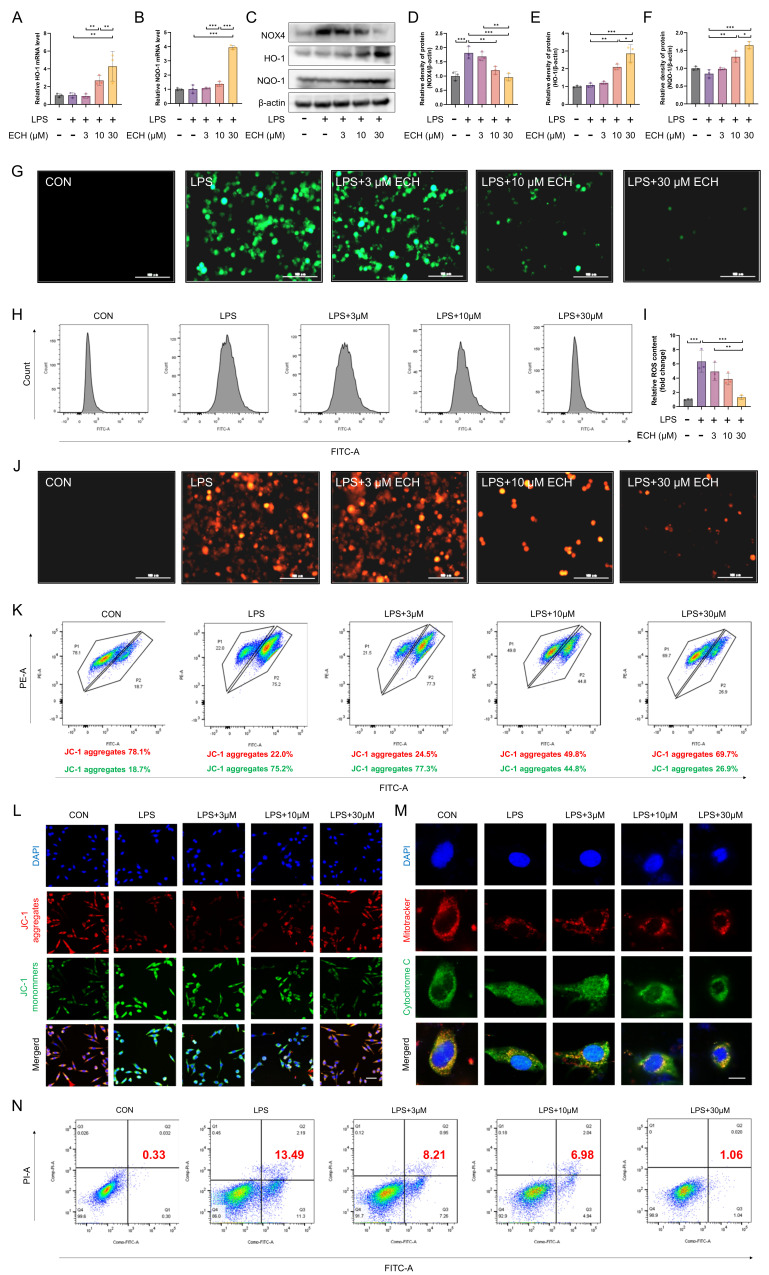
Echinacoside protected endothelial cells from mitochondrial pathway apoptosis. (**A**–**F**) mRNA levels of HO-1 and NQO-1, and protein levels of HO-1, NQO-1, and NOX4 were measured using qPCR and Western blotting. (**G**–**I**) ROS levels labeled with H2DCFCDA were detected using flow cytometry and under a microscope. Bars represent 100 μm. The microscope’s magnifications are the same. (**J**) A Mitosox probe was used to label mitochondrial ROS levels and was detected under a microscope. Bars represent 100 μm. The microscope’s magnifications are the same. (**K**,**L**) JC-1 staining was performed to assess mitochondrial membrane potential using flow cytometry and under a microscope. Bars represent 50 μm. The microscope’s magnifications are the same. (**M**) Immunofluorescence staining showed the co-staining of Mitotracker (red) and Cytochrome C (green), which was observed under a microscope. Bars represent 20 μm. The microscope’s magnifications are the same. (**N**) Endothelial cell apoptosis was assessed using flow cytometry. The data are presented as mean ± standard deviation, and all experiments were repeated independently at least three times. (* *p* < 0.05, ** *p* < 0.01, *** *p* < 0.001).

**Figure 5 antioxidants-12-01925-f005:**
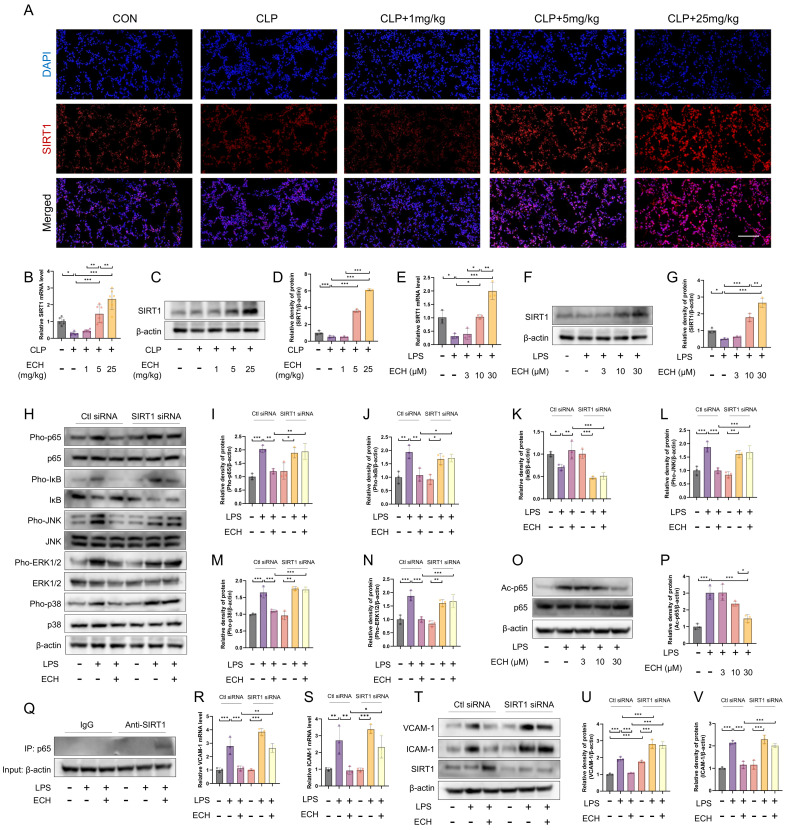
SIRT1 mediated the effect of echinacoside on the pathological activation of endothelial cells. (**A**–**D**) Immunofluorescence was used to assess SIRT1 levels in lung tissue sections. The mRNA and protein levels of SIRT1 in lung tissue homogenates were measured using qPCR and Western blotting, respectively. Bars represent 100 μm. The microscope’s magnifications are the same. (**E**–**G**) The mRNA and protein levels of SIRT1 in endothelial cells were measured using qPCR and Western blotting. (**H**–**N**) Under Control siRNA and SIRT1 siRNA treatments, the protein levels of Pho-p65, p65, Pho-IκB, IκB, Pho-ERK1/2, ERK1/2, Pho-JNK, JNK, Pho-p38, and p38 were assessed by Western blotting. (**O**,**P**) The protein levels of Ac-p65 and p65 were measured using Western blotting. (**Q**) The binding of p65 and SIRT1 was evaluated using CO-IP. (**R**–**V**) Under Control siRNA and SIRT1 siRNA treatments, the mRNA and protein levels of VCAM-1 and ICAM-1 were measured using qPCR and Western blotting. The data are presented as mean ± standard deviation, and all experiments were repeated independently at least three times. (* *p* < 0.05, ** *p* < 0.01, *** *p* < 0.001).

**Figure 6 antioxidants-12-01925-f006:**
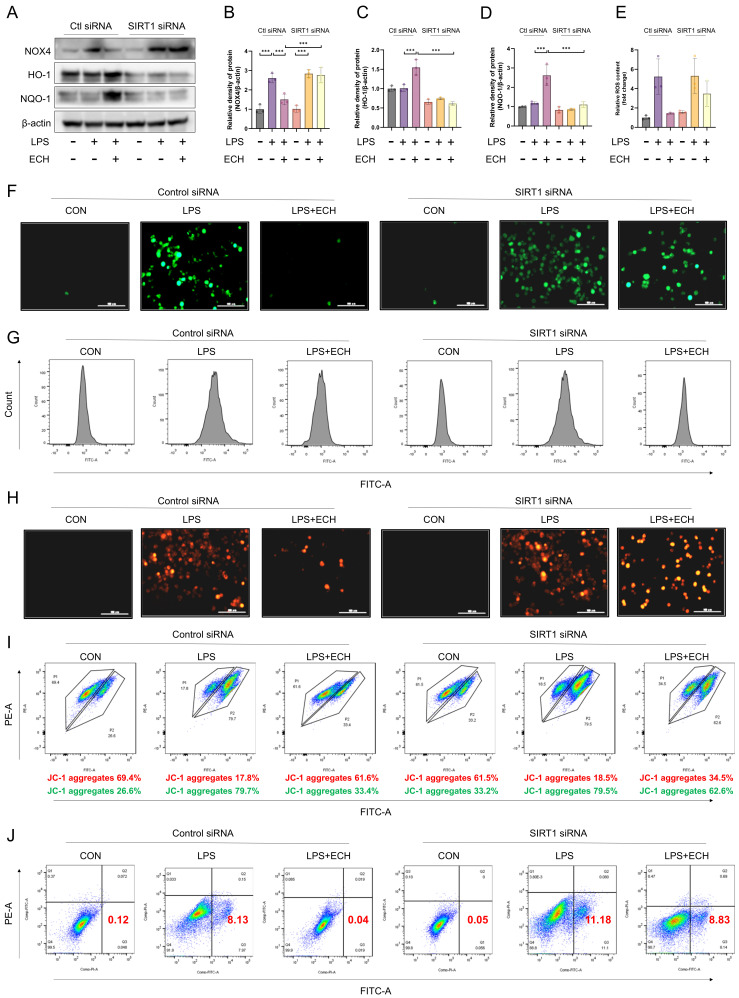
SIRT1 mediated the protective effect of echinacoside on endothelial cell apoptosis via the mitochondrial pathway. (**A**–**D**) Under Control siRNA and SIRT1 siRNA treatments, the protein levels of HO-1, NQO-1, and NOX4 were assessed by Western blotting. (**E**–**G**) Under Control siRNA and SIRT1 siRNA treatments, the levels of ROS labeled with H2DCFCDA were detected using flow cytometry and observed under a microscope. Bars represent 100 μm. The microscope’s magnifications are the same. (**H**) Under Control siRNA and SIRT1 siRNA treatments, the mitochondrial ROS levels labeled with a Mitosox probe were quantified and observed under a microscope. Bars represent 100 μm. The microscope’s magnifications are the same. (**I**) Under Control siRNA and SIRT1 siRNA treatments, JC-1 staining was performed to assess mitochondrial membrane potential using flow cytometry. Bars represent 50 μm. (**J**) Under Control siRNA and SIRT1 siRNA treatments, endothelial cell apoptosis was assessed using flow cytometry. The data are presented as mean ± standard deviation, and all experiments were repeated independently at least three times. *** *p* < 0.001).

**Figure 7 antioxidants-12-01925-f007:**
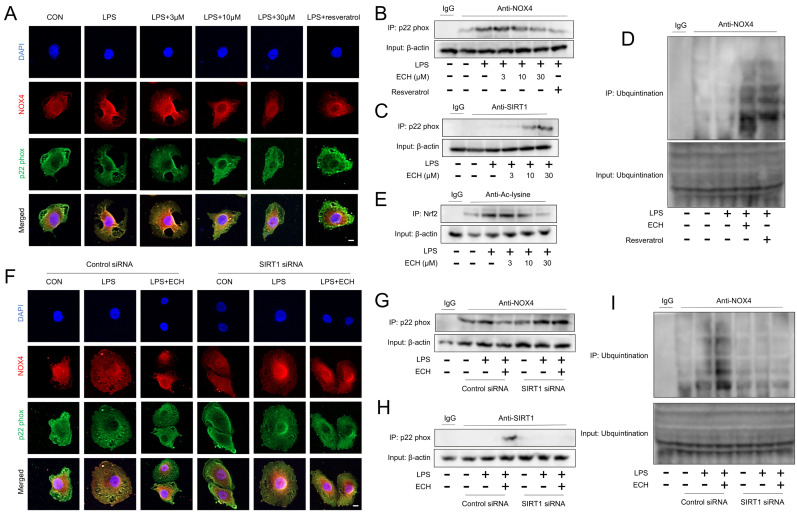
Activated SIRT1 inhibited NOX4 activation and promoted NOX4 ubiquitination degradation. (**A**) Endothelial cells were treated with different concentrations of ECH and SIRT1 activator resveratrol. Immunofluorescence was used to assess the co-staining of NOX4 (red) and p22 phox (green). Bars represent 20 μm. The microscope’s magnifications are the same. (**B**) CO-IP was performed to assess the binding of NOX4 and p22 phox. (**C**) CO-IP was performed to assess the binding of SIRT1 and p22 phox. (**D**) CO-IP was used to measure the ubiquitination level of NOX4. (**E**) CO-IP was used to measure the acetylation level of Nrf2. (**F**) Under Control siRNA and SIRT1 siRNA treatments, immunofluorescence was used to assess the co-staining of NOX4 (red) and p22 phox (green). Bars represent 20 μm. The microscope’s magnifications are the same. (**G**) Under Control siRNA and SIRT1 siRNA treatments, CO-IP was performed to assess the binding of NOX4 and p22 phox. (**H**) Under Control siRNA and SIRT1 siRNA treatments, CO-IP was performed to assess the binding of SIRT1 and p22 phox. (**I**) Under Control siRNA and SIRT1 siRNA treatments, CO-IP was used to measure the ubiquitination level of NOX4. The data are presented as mean ± standard deviation, and all experiments were repeated independently at least three times.

**Figure 8 antioxidants-12-01925-f008:**
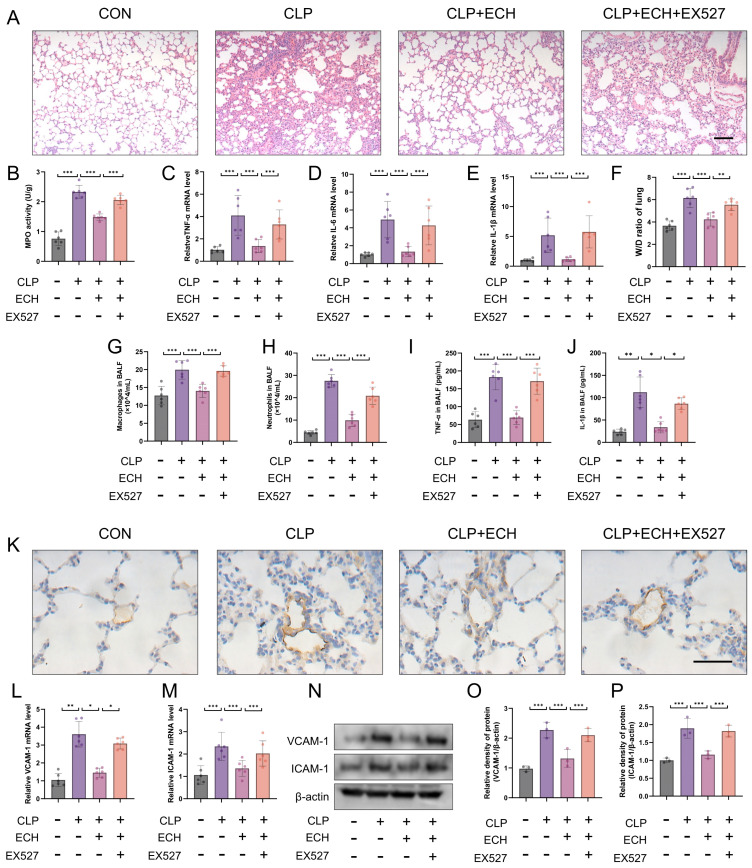
SIRT1 mediated the protective effect of echinacoside on reducing sepsis-induced acute lung injury and preserving endothelial cell function. (**A**) Lung morphology was assessed using HE staining. Bars represent 100 μm. The microscope’s magnifications are the same. (**B**) MPO levels were measured using biochemical methods. (**C**–**E**) The mRNA levels of TNF-α, IL-6, and IL-1β in lung tissue homogenates were quantified using qPCR. (**F**) The lung W/D ratio was measured. (**G**,**H**) The content of macrophages and neutrophils in BALF was determined using Giemsa staining. (**I**,**J**) The levels of TNF-α and IL-Iβ in BALF were measured using ELISA. (**K**) The level of VCAM-1 in lung tissue was evaluated using immunohistochemistry. Bars represent 100 μm. The microscope’s magnifications are the same. (**L**–**P**) The mRNA and protein levels of VCAM-1 and ICAM-1 were quantified using qPCR and Western blotting, respectively. The data are presented as mean ± standard deviation, and all experiments were repeated independently at least three times. (* *p* < 0.05, ** *p* < 0.01, *** *p* < 0.001).

**Figure 9 antioxidants-12-01925-f009:**
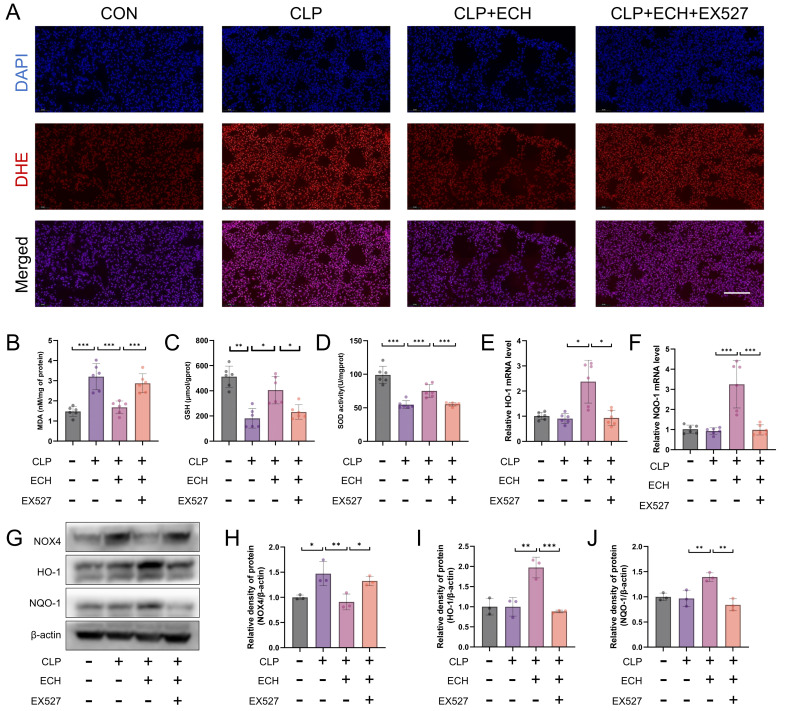
SIRT1 mediated the effect of echinacoside in alleviating sepsis-induced lung oxidative stress. (**A**) DHE staining reflects the level of pulmonary oxidative stress. Bars represent 100 μm. The microscope’s magnifications are the same. (**B**–**D**) The levels of MDA, GSH, and SOD were measured using biochemical methods. (**E**–**J**) The mRNA levels of HO-1 and NQO-1 and the protein levels of HO-1, NQO-1, and NOX4 were quantified using quantitative PCR and Western blotting, respectively. The data are presented as mean ± standard deviation, and all experiments were repeated independently at least three times. (* *p* < 0.05, ** *p* < 0.01, *** *p* < 0.001).

## Data Availability

No new data were created or analyzed in this study. Data sharing is not applicable to this article.

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
