# Peer review of "Sirtuin1 Mediates the Protective Effects of Echinacoside against Sepsis-Induced Acute Lung Injury via Regulating the NOX4-Nrf2 Axis"

_antioxidants, 2023, doi:10.3390/antiox12111925_

Round 1

Reviewer 1 Report

In this paper, the authors explore the effects of echinacoside on sepsis-induced acute lung injury and oxidative stress in mice. Additionally, they also study the regulatory mechanisms of Sirtuin1 on NOX4. The paper has a lot of content, and the conclusions about the positive effect of echinacoside are unequivocal, given the results. Nevertheless, in my opinion, the paper can be noticeably improved by helping readers tackle the considerable amount of results and by clearing some important issues regarding statistics.

- First, no exact p-values are given. Results of the statistical analyses are only presented in figures and only when they are statistically significant. All p-values should be reported (significant and non-significant) at least to the nearest thousandth (or hundredth if the p-value is large).

- If I haven't miscounted, the authors have performed around 400 statistical tests. It's hard to follow all these results, so maybe they could add a summary table with all the comparisons performed and their results to have an overall picture.

- In the methods section, the authors report that ANOVA was used to assess differences among groups, followed by the Student-Newman-Kersee (SNK) post hoc test. Were the main assumptions for ANOVA (equal variances and normality) assessed? Also, I had never heard about Student-Newman-Kersee before. SNK usually stands for Student-Newman-Keuls. Are the authors talking about this method? If so, I do not think it is appropriate to use this method since it fails to properly control the type I error rate. They should have used a more standard method, such as Tukey's method. If they want to keep the SNK results, they should, at least, perform a sensitivity analysis using Tukey's method to show the robustness of their results. 

Author Response

I apologize for the confusion. We have revised and marked in red in the manuscript (lines 268-275). And all the data analysis results have been added to supplementary material 1.

All data were expressed as means ± standard deviations and analyzed using GraphPad Prism 9.0 software. Firstly, we performed a Shapiro-Wilk test to assess the normality of the data distribution. For data that followed a normal distribution, one-way analysis of variance (ANOVA) was used for intergroup statistical comparisons, followed by the Turkey test for further pairwise group comparisons. For data that did not follow a normal distribution, we employed the Kruskal-Wallis test. Survival analysis was conducted using the log-rank test. Statistical significance was considered when the p-value was less than 0.05.

Reviewer 2 Report

The authors present data suggesting that SIRT1 mediates the antioxidant effects of echinacoside in a work in which they have studied sepsis-induced acute lung injury. They report that echinacoside inhibits sepsis-induced acute lung injury and oxidative stress. Therefore, in this current work echinacoside mediates antioxidative effect (reduces ROS). In previous works from others echinacoside has been used in cancer therapy, e.g., PMID:35734366 PMID:26677335 PMID:26132569, in which it has increased ROS levels.

Explain the contradictory data. Is it because of the dose of echinacoside? If so, what is the mechanism? In the work PMID:35734366 the authors showed that echinacoside inhibits PI3K/AKT pathway. This is in line with reduced NADPH oxidase NOX2 activity and reduced superoxide anion production but the overall biological outcome of echinacoside is too confusing. The reader cannot believe the data if you do not explain the contradictory results.

Author Response

Thank you very much for your questions. In response to the issues raised, we have conducted a series of additional experiments (Supplementary Fig. 1) and have incorporated these results into our manuscript (lines 572-585, 615-631). Here, we provide a comprehensive elucidation of these experimental results:

The reviewer mentioned that in some studies, such as PMID:35734366, treatment with echinacoside could inhibit the AKT signaling pathway, increase intracellular ROS levels, and promote apoptosis in endometrial cancer cells (Ishikawa and HEC-1-B cells). This study's findings and conclusions regarding the pharmacological activity of echinacoside, especially in regulating ROS at the intracellular level, differ somewhat from those in the mentioned study. To explore the reasons behind these differences, we treated HUVECs with the drug concentrations used in that study (PMID:35734366, approximately 200 µg/mL, which is roughly 255 µM) and the effective drug concentration used in our study (30 µM). We observed the impact of echinacoside at different concentrations on intracellular ROS and conducted a preliminary exploration of its mechanisms.

The research results indicate that, compared to the control group, a high concentration of echinacoside (255 µM) significantly increased the intracellular ROS content in HUVECS and promoted cell apoptosis, while treatment with a low concentration of echinacoside (30 µM) showed no significant changes (Supplementary Fig. 1A, C). Subsequently, we further explored the mechanisms behind this phenomenon. The study revealed that 255 µM echinacoside could alter mitochondrial membrane potential, leading to an increase in MitoROS levels (Supplementary Fig. 1 B, D). Therefore, we believe that the mechanism by which high concentrations of echinacoside increase intracellular ROS and promote apoptosis may be related to its induction of mitochondrial damage.

  Given that the AKT signaling pathway can also influence the activity of NADPH oxidase, we separately observed the impact of different concentrations of echinacoside on the AKT signaling pathway. The study revealed that a high concentration of echinacoside could significantly inhibit the activation of AKT, while this phenomenon was not observed with low-concentration echinacoside treatment (Supplementary Fig. 1E). Therefore, we believe that the inhibitory effect of echinacoside on the AKT signaling pathway is closely related to its concentration.

In this study, echinacoside was used at 30 µM. At this concentration, echinacoside significantly enhanced the activity of Sirt1 while having no significant impact on the AKT signaling pathway. Furthermore, in conjunction with the previous research results, it was observed that knocking down Sirt1 expression led to a notable decrease in echinacoside's antioxidative effects (Fig. 6F-H). We believe that Sirt1 mediates the biological effects of echinacoside at low concentrations. However, further investigation is required to understand the effects of high concentrations of echinacoside on Sirt1.

  It's worth mentioning that the Sirt1 activator Resveratrol also exhibits similar pharmacological activities. Its impact on cellular oxidative stress response is also bidirectional, meaning that at low concentrations, it exerts antioxidative stress/cellular protective effects, while at high concentrations, it promotes oxidative stress/cell toxicity[1].

[1] Shaito A, Posadino AM, Younes N, et al. Potential Adverse Effects of Resveratrol: A Literature Review. Int J Mol Sci. 2020 Mar 18;21(6):2084. doi: 10.3390/ijms21062084.

Round 2

Reviewer 1 Report

The authors have addressed all the issues raised in the previous review. Just one minor comment: The correct spelling for the posthoc test is "Tukey", not "Turkey".

Also, the fact that the authors performed so many statistical analyses impacts the proportion of false positives. Given all the presented results, it is evident that echinacoside has a positive effect. However, given the sheer amount of tests performed, even after controlling the Family Wise Error Rate at 0.05 for each ANOVA, the probability of, at least, having one false positive in all the presented results is large, and this should be included in the limitations of the study.

Author Response

Thank you very much for your valuable comments and questions. For your questions, we will make the following answers.

Q: T The authors have addressed all the issues raised in the previous review. Just one minor comment: The correct spelling for the posthoc test is "Tukey", not "Turkey".

Also, the fact that the authors performed so many statistical analyses impacts the proportion of false positives. Given all the presented results, it is evident that echinacoside has a positive effect. However, given the sheer amount of tests performed, even after controlling the Family Wise Error Rate at 0.05 for each ANOVA, the probability of, at least, having one false positive in all the presented results is large, and this should be included in the limitations of the study.

A: I apologize for the confusion. We have revised and marked in red in the manuscript (line 274). And we have added a discussion of the limitations of the statistical methods in our manuscript (lines 659-663).

Reviewer 2 Report

N/A

Author Response

Thank you very much for your valuable review comments!